

# Ammonia emission estimates using CrIS satellite observations over Europe

Jieying Ding[1*], Ronald van der A[1], Henk Eskes[1], Enrico Dammers[2], Mark Shephard[3], Roy Wichink Kruit[4], Marc Guevara[5], Leonor Tarrason[6]

1. Royal Netherlands Meteorological Institute (KNMI), De Bilt, The Netherlands

2. Netherlands Organisation for Applied Scientific Research (TNO), Utrecht, The Netherlands

3. Environment and Climate Change Canada (ECCC), Toronto, Ontario, Canada

4. National Institute for Public Health and the Environment, Bilthoven, The Netherlands

5. Barcelona Supercomputing Center, Barcelona, Spain

6. NILU – Norwegian Institute for Air Research, Kjeller, Norway

*Corresponding authors: Jieying Ding (jieying.ding@knmi.nl)

Abstract

Over the past century ammonia (NH$_3$) emissions have increased with the growth of livestock and fertilizer usage. The abundant NH$_3$ emissions lead to secondary fine particulate matter (PM2.5) pollution, climate change, reduction in biodiversity and affects human health. Up-to-date and spatially and temporally resolved information of NH$_3$ emissions is essential to better quantify its impact. In this study we applied the existing DECSO (Daily Emissions Constrained by Satellite Observations) algorithm to NH$_3$ observations from the Cross-track Infrared Sounder (CrIS) to estimate NH$_3$ emissions. Because NH$_3$ in the atmosphere is influenced by Nitrogen Oxides (NO$_x$), we implemented DECSO to estimate NO$_x$ and NH$_3$ emissions simultaneously. The emissions are derived over Europe for 2020 on a spatial resolution of 0.2° × 0.2° using daily observations from both CrIS and TROPOMI (on the Sentinel 5p satellite). Due to the sparseness of daily satellite observations of NH$_3$, monthly emissions of NH$_3$ are reported. The total NH$_3$ emissions derived from observations are about 8 Tg/year with a precision of about 0.2 % over the European domain. The comparison of the satellite-derived NH$_3$ emissions from DECSO with independent bottom-up inventories and in-situ observations indicates a consistency in



terms of magnitude on the country totals, the results also being comparable regarding the temporal and
spatial distributions.


1 Introduction

Ammonia ($NH_3$) is the most abundant alkaline gas and one of the main reactive nitrogen species in the
atmosphere. $NH_3$ is a precursor for the formation of atmospheric aerosols, which play an important role
in climate change. In Europe, about 50% (Wyer et al., 2022) of atmospheric $NH_3$ is transformed into
fine particulate matter (PM2.5) composed of ammonium through chemical reactions with sulfuric and
nitric acids from nitrogen oxides ($NO_x$) and sulphur dioxides ($SO_2$) in the atmosphere (Renard et al.,
2004; Schaap et al., 2004). According to the European Environment Agency (EEA), the dominant
source of $NH_3$ in Europe is agriculture, which was responsible for more than 90% of the European
emissions. The other source sectors include industry, transport, energy, waste treatment and biomass
burning (Behera et al., 2013; Backes et al., 2016a; Van Damme et al., 2018; Adams et al., 2019).
Excessive $NH_3$ emissions have adverse impact on biodiversity, human health, and climate change
(Galloway et al., 2008). Over the past century, $NH_3$ emissions increased strongly with the growing
human population, cattle farming and fertilizer usage (Crippa et al., 2023; Erisman et al., 2008; Van
Damme et al., 2021), leading to high nitrogen deposition loads to water and soil (Erisman et al., 2013)
with the associated eutrophication, acidification and biodiversity loss problems (Behera et al., 2013).
Since 2019, the Dutch policy makers paid a lot of attention to $NH_3$ emissions due to the nitrogen (N)
crisis after the national programmatic approach to nitrogen was rejected by the supreme court, because
it was inadequate for the protection of vulnerable nature areas (Natura2000). The Dutch government is
obliged by EU laws to protect the natural environment and prevent damage caused by too high
emissions of reactive nitrogen. Studies shows that abatement of $NH_3$ emissions is very cost-effective to
improve air quality and have high social benefits (Backes et al., 2016b; Zhang et al., 2020; Gu et al.,
2021). Detailed spatially and temporally resolved information of $NH_3$ emissions is crucial for both
scientific communities and policy makers to study and predict pollutant concentrations and deposition
with their impact on the environment and to motivate environmental control strategies.
The empirical method to estimate $NH_3$ emissions is the so-called bottom-up approach, which combines
available official reported activity data incorporating a full differentiation of emission activities with
emission factors, and technology and abatement measures from individual countries for each source
category (Crippa et al., 2018; Crippa et al., 2023; Janssens-Maenhout et al., 2019). The annual emissions
are then distributed in time and space based on proxy data such as land use data, and meteorological



parameters (Backes et al., 2016a). Ge et al. (2020) summarized the key factors of agricultural NH$_3$
emissions: local agricultural practices, method of manure and fertilizer application including type,
amount and method, animal type, housing type, manure storage type, meteorological conditions, soil
conditions, and regulation of agricultural practice. The uncertainties of NH$_3$ emissions calculated by the
bottom-up approach are very large due to insufficient data on agricultural activities (Behera et al., 2013;
Beusen et al., 2008). Crippa et al. (2018) pointed out that the uncertainty of NH$_3$ (between 186 % and
294.4 %) in the EDGAR (The Emissions Database for Global Atmospheric Research) inventory is the
largest among all pollutants because of the high uncertainty of both agricultural statistics and emission
factors.
The validation of NH$_3$ emission inventories using ground-based observations is very challenging due
to the sparsely distributed in-site measurement network. NH$_3$ concentrations have large temporal and
spatial variability due to its short lifetime, which ranges from about a few hours to two days (Dammers
et al., 2019; Luo et al., 2022).  Densely distributed hourly or daily ground measurements are impractical
for large areas due to high costs and specific operational requirements (Noordijk et al., 2020). In the
last decade, a wide spatial and temporal coverage of satellite observations of NH$_3$ in lower troposphere
was established due to the development of infrared nadir viewing satellite instruments, such as the
Tropospheric Emission Spectrometer (TES) (Beer et al., 2008) on the NASA Aura satellite. The
operational Cross-track Infrared Sounder (CrIS) (Shephard and Cady-Pereira, 2015) on the Suomi
National Polar-orbiting Partnership (S-NPP) and on the Joint Polar Satellite System-1 and System-2
(JPSS-1 and JPSS-2) satellites of NASA/NOAA, and the Infrared Atmospheric Sounder Interferometer
(IASI) (Clarisse et al., 2009) on the MetOp satellites from the European Space Agency (ESA), with
their large swaths, provide daily global coverage of NH$_3$ observations and improve our understanding
of NH$_3$ global distribution and temporal variability.
NH$_3$ emissions can be obtained by applying an inversion algorithm to satellite observations. Such
estimates provide useful information which is independent from bottom-up inventories. By using IASI
NH$_3$ observations, Van Damme et al. (2018) identified NH$_3$ emission hotspots and calculated emissions
based on a mass balance approach. They found that NH$_3$ emissions of most hotspots, especially
industrial emitters, were largely underestimated compared to EDGAR. Dammers et al. (2019) used both
IASI and CrIS observations to derive emissions, lifetimes and plume widths of NH$_3$ from large
agricultural and industrial point sources and concluded that 55 locations were missing in the
Hemispheric Transport Atmospheric Pollution version 2 (HTAPv2) emission inventory. Besides the
studies on point sources, data assimilation techniques combining a chemical transport model (CTM)
with satellite observations are also widely used to derive NH$_3$ surface emissions. van der Graaf et al.
(2022) adjusted the NH$_3$ emissions over Europe using a local ensemble transport Kalman filter (LETKF)
applied to CrIS NH$_3$ profiles. Sitwell et al. (2022) developed an ensemble-variational inversion system
to estimate NH$_3$ emissions from CrIS over North America. Another widely used method is 4D-Var



using the GEOS-Chem global chemistry transport model, which has been applied to America, China
and Europe using $NH_3$ observation from different instruments (Zhu et al., 2013; Zhang et al., 2018; Li
et al., 2019; Cao et al., 2020; Chen et al., 2021; Cao et al., 2022). The main advantage of CrIS is the
combination of global coverage and the improved sensitivity in the boundary layer attributed to the low
spectral noise of about 0.04 K at 280 K in the $NH_3$ spectral band (Zavyalov et al., 2013). The infrared
instrument is also more sensitive at the overpass time in the early afternoon with high thermal contrast
between air and surface.
The Daily Emissions Constrained by Satellite Observations (DECSO) inversion algorithm uses satellite
column observations to derive emissions for short-lived gases based on an extended Kalman Filter
(Mijling and van der A, 2012).  The concentrations of the species are calculated from the emissions by
a CTM and compared to satellite observations. One of the main advantages to use DECSO is the fast
calculation speed compared to other data assimilation methods. Furthermore, the derived emissions are
updated by addition, not by scaling the existing emissions. This enables the fast detection of new sources
and changed emissions. In previous studies, DECSO has been applied to nitrogen dioxide ($NO_2$)
observations from different satellites and uses the Eulerian regional off-line CTM CHIMERE (Menut
et al., 2021; Menut et al., 2013) to estimate regional NOx ($NO_2$+NO) emissions and it revealed that the
temporal and spatial variability of total surface $NO_x$ emissions are well captured by DECSO compared
to bottom-up inventories or in-situ observations (Ding et al., 2015; Ding et al., 2017b; Ding et al., 2020;
van der A et al., 2020; Ding et al., 2022; Liu et al., 2018; van der A et al., 2024).
Direct validation of emission inventories, regardless of bottom-up or satellite-derived approaches,
presents the same challenge due to the inherent difficulty of directly measuring large-scale emissions
on the ground. The intercomparison of emissions using independent data and different approaches are
usually performed to assess the emission data. Another common way to validation emissions can be
achieved by using them as input data in a chemical transport model. The model simulated concentrations
are compared to in-situ observations.
In this study we extend the DECSO-NOx system to $NH_3$ in order to derive both $NO_x$ and $NH_3$ emissions
simultaneously, using CrIS $NH_3$ observations and $NO_2$ observations from the TROPOspheric
Monitoring Instrument (TROPOMI) (Veefkind et al., 2012). Using the multi-species DECSO version,
we update $NO_x$ and $NH_3$ emissions simultaneously to reduce the impact of the temporal change (e.g.
trend) of $NO_x$ when deriving $NH_3$ emissions. After the description of the DECSO algorithm applied to
$NH_3$, the results of $NH_3$ emissions over Europe are presented at a spatial resolution of $0.2° \times 0.2°$. To
evaluate the derived $NH_3$ emissions, we will compare the country totals and the monthly variability
with bottom-up inventories with a focus on $NH_3$ emissions in the Netherlands. In addition, we compare
the $NH_3$ concentration simulations of CHIMERE using different emission inventories with in-situ
observations.




## 2 Data and Method

### 2.1 Satellite observations

#### *2.1.1 CrIS observations of NH$_3$*

The CrIS instrument is a Fourier transform spectrometer (FTS) launched on the Suomi National Polar-orbiting Partnership (SNPP) satellite in 2011 and on the NOAA-20 satellite in 2017. The overpass time of SNPP at the equator is about 01:30 and 13:30 local time. NOAA-20 circles the earth in the same orbit as SNPP, but it is separated in time and space by 50 minutes and crosses the equator at about 02:20 and 14:20 local time. The instrument has a wide swath of up to 2200 km providing twice daily global coverage. The total angular field-of-view consists of a 3×3 array of circular pixels of 14 km diameter each at nadir (Han et al., 2013). CrIS measures the infrared spectrum including the main NH$_3$ spectral signatures located in the longwave window region between 900 and 1000 cm$^{-1}$. The spectral resolution of the radiance data is 0.625 cm$^{-1}$. NH$_3$ observations are retrieved with the CrIS Fast Physical Retrieval (CFPR) algorithm based on an optimal estimate method minimizing the difference between measured spectral radiances and those simulated by a radiative transfer model (Shephard and Cady-Pereira, 2015). Three typical a priori profiles of NH$_3$ representing high-source, moderate-source and background source are used in the retrieval algorithm. The NH$_3$ profile are retrieved on 14 pressure levels with the peak sensitivity of CrIS between 900 and 700 hPa (Shephard et al., 2020). We use the version 1.6.4 retrieval products of CrIS on both SNPP and NOAA-20 from September 2019 to December 2020, which also accounts for non-detects in the observations and retrievals through optically thin clouds (White et al., 2023). We use the daytime observations with the quality flag larger than 3 over our study domain of Europe [-10° ~30° E, 35° ~ 55° N] (Shephard et al., 2020). Since there are almost no emissions over ocean, we only use the observations over land. To reduce extreme emission updates in one day we filter the NH$_3$ data larger than the value at 99[th] percentile of all observations for the selected period over the study domain. This has also been applied by van der Graaf et al. (2022). To make a fair comparison between NH$_3$ observations of CrIS and model simulations of CHIMERE, we interpolate modelled concentrations from the model grid cell over the satellite footprints and apply the averaging kernel to the modelled profile. Although the NH$_3$ observations from CrIS are in circular pixels, we still assume the pixel to be rectangular and calculate the pixel corner coordinates based on the satellite height, satellite zenith angle and viewing angle assuming the width of the pixel to be equal to the diameter of the circular pixel. To simplify the calculation of applying the original logarithmic averaging kernels, we converted them to linearized average kernels based on the method of Cao et al. (2022).

167



*2.1.2 TROPOMI observations of NO₂*

TROPOMI is onboard the Sentinel-5 Precursor (S5P) satellite launched on 13 October 2017 with the high spatial resolution of $3.5 \times 5.5$ km$^2$ at nadir for the NO$_2$ observations. The overpass time is about 13:30 local time, similar as for CrIS. We use TROPOMI tropospheric NO$_2$ columns from the version 2.4 reprocessed retrieval dataset (van Geffen et al., 2022) and follow the recommendations for using the QA value as detailed in the Product User Manual (Eskes and Eichmann, 2022). NO$_2$ columns are converted into 'superobservations' representing the integrated average (Boersma et al., 2016; Rijsdijk et al., 2024) over the $0.2° \times 0.2°$ grid cells. In this paper, the superobservations are calculated for the NO$_2$ columns from surface till about 700hPa where the NO$_2$ concentrations are most related to surface emissions. The details of TROPOMI NO$_2$ data used by DECSO are described in Ding et al. (2020) and van der A et al. (2024).

2.2 Ground-based observations.

To evaluate the NH$_3$ emissions derived by DECSO, we use independent ground-based observations in 2020 to compare with model simulated NH$_3$ concentrations of CHIMERE using different inventories. Compared to other countries, Netherlands has the densest network for monitoring surface NH$_3$ concentrations. We use hourly NH$_3$ concentrations measured by mini-DOAS at six locations (Figure S1) from the Dutch Monitoring Air Quality (LML) network (Berkhout et al., 2017) and monthly measurements of NH$_3$ concentration provided by passive samples at 394 locations (Figure S2) from the Dutch Measuring Ammonia in Nature (MAN) network (Lolkema et al., 2015). The uncertainty in NH$_3$ concentrations measured with individual passive samples is large and the measurements are calibrated monthly against the high-quality measurements from the LML network to enhance the accuracy.

2.3 Emission inventories

To verify the satellite-derived emissions of NH$_3$ in Europe, we compare them to several emission inventories including: the national emissions inventories officially reported under the Convention on Long-range Transboundary Air Pollution (LRTAP) (Pinterits, 2023) of 2020, the emissions reported under the European Pollutant Release and Transfer Register (E-PRTR) (EPRTR, 2012) of 2020 including releases from industrial facilities and livestock facilities, the global emission inventory Hemispheric Transport of Air Pollution (HTAP) v3 of 2018 (Crippa et al., 2023), the Copernicus Atmosphere Monitoring Service (CAMS) Global anthropogenic emissions (CAMS-GLOB-ANT) v5.3 of 2020 (Soulie et al., 2023), the regional European CAMS anthropogenic emission inventory (CAMS-REG-ANT) v5.1 of 2020 (Kuenen et al., 2022) and the Dutch official registered emissions of NH$_3$ in 2020 (https://data.emissieregistratie.nl/export) (see Table 1). HTAP v3 has been developed by



integrating official inventories over specific areas including CAMS-REG-ANT v5.1 for Europe with
the EDGAR v6.1 inventory for the remaining world regions with the spatial resolution of 0.1°× 0.1°.
CAMS-GLOB-ANT combines the EDGAR annual emissions and the Copernicus Atmosphere
Monitoring Service TEMPOral profiles (CAMS-TEMPO) on a global scale (Guevara et al., 2021). The
emissions of the most recent years are calculated based on the trends from the Community Emissions
Data System (CEDS) global inventory (Hoesly et al., 2018). The resolution of CAMS-GLOB-ANT is
0.1°× 0.1°.  CAMS-REG-ANT v5.1 provide yearly emissions on the spatial resolution of 0.1°× 0.05°.
We have applied the regional European CAMS-TEMPO profiles (Guevara et al., 2021) to CAMS-REG-
ANT v5.1 to get the monthly emissions (hereinafter referred to as CAMS-REG-TEMPO). The Dutch
registered $NH_3$ emissions are taken from https://www.emissieregistratie.nl and provided annually on a
high resolution of 1 km ×1 km. To compare the derived $NH_3$ emissions of DECSO spatially with
bottom-up inventories, we aggregate emissions from these bottom-up inventories into the 0.2° × 0.2°
grid cells of the DECSO working domain.

*Table 1. Summary of the bottom-up inventories compared to the satellite-derived $NH_3$ emissions from DECSO.*

| Emission inventory | Year | Spatial Resolution | Temporal resolution |
|---|---|---|---|
| LRTAP | 2020 | Country total | Annual |
| E-PRTR | 2020 | Point source | Annual |
| HTAP v3 | 2018 | 0.1°× 0.1° | Monthly |
| CAMS-GLOB-ANT v5.3 | 2020 | 0.1°× 0.1° | Monthly |
| CAMS-REG-ANT v5.1 | 2020 | 0.1°× 0.05° | Annual, monthly (with CAMS-REG-TEMPO) |
| Dutch Registered $NH_3$ emissions | 2020 | 1 km ×1 km | Annual |


2.4 DECSO
DECSO is an inversion algorithm developed for the purpose of deriving emissions of short-lived species
from satellite observations. As such DECSO has been specifically designed to use daily satellite
observations of column concentrations to provide rapid updates of emission estimates of short-lived
atmospheric constituents on a regional scale. An extended Kalman filter is used, in which emissions are
translated to column concentrations via the CTM and these are compared to the satellite column

  



observations. Based on that single forward CTM simulation, the sensitivity of concentrations to
emissions is calculated by using trajectory analyses to account for transport away from the source. In
previous studies, DECSO has been applied to $NO_2$ observations from different satellites including
TROPOMI to estimate $NO_x$ emissions (Mijling et al., 2013; Ding et al., 2015; van der A et al., 2020;
Ding et al., 2022; Ding et al., 2020; van der A et al., 2024). The studies revealed that the temporal and
spatial variability of total surface $NO_x$ emissions are captured well by DECSO (Ding et al., 2017b; van
der A et al., 2017; Liu et al., 2018). Here we have used the updated version DECSO v6.3 (van der A et
al., 2024) for estimating simultaneously $NO_x$ and $NH_3$ emissions using the daily observations from
TROPOMI and CrIS (referred to as multi-species DECSO). The main changes of v6.3 include
improving the sensitivity matrix calculation and using the latest Eulerian regional off-line CTM
CHIMERE v2020v3 (Menut et al., 2021) instead of CHIMERE v2013. In the CTM, we employ the
Copernicus Landcover 2019 data (Buchhorn et al., 2020) , and the source sector distributions of
emissions obtained from HTAP v3 of 2018, which are also used as input emissions of other species
beside $NO_x$ and $NH_3$. CHIMERE is driven by the operational meteorological forecast of the European
Centre for Medium-Range Weather Forecasts (ECMWF). Here we present the specific setting in
DECSO for $NH_3$ (referred to as DECSO-NH3).
To update $NH_3$ emissions based on the Kalman filter equations, one of the essential calculations is the
Kalman gain matrix (**K**) using the following equation:
$$\mathbf{K} = \mathbf{P}^f(t)\mathbf{H}[\mathbf{H}\mathbf{P}^f(t)\mathbf{H}^T + \mathbf{R}]^{-1} \qquad\qquad (1)$$
$\mathbf{P}^f$ is the error covariance matrix of the forecasted emissions at time $t$. **H** is the sensitivity matrix
(Jacobian) describing how the $NH_3$ column concentration on a satellite footprint depends on gridded
$NH_3$ emissions. **R** is the error covariance combining the observation error of tropospheric $NH_3$ columns,
the uncertainty of the CTM, and representation error introduced by projection of modelled columns on
the satellite footprint.
$\mathbf{P}^f$ is parametrised based on an evaluation of the emission forecast error $q$, which is the error increase
during one time step of the forecast model. The emission forecast model is persistence, predicting that
the emission is equal to the analysis of the emissions from the previous day. We parametrize $q$ of $NH_3$
following:
$$q = \varepsilon_{abs} \exp\left(-\frac{\varepsilon_{rel}}{\varepsilon_{abs}} e\right) + \varepsilon_{rel} e \qquad\qquad (2)$$
$\varepsilon_{abs}$ and $\varepsilon_{rel}$ are the absolute and relative errors that are the dominating emission errors for low and high
emissions respectively.
To determine $\varepsilon_{abs}$, $\varepsilon_{rel}$ and also the covariance matrix **R** for $NH_3$, we follow the method described by
Ding et al. (2017a) based on the analysis of Observation minus Forecast (OmF) and Observation minus



Assimilation (OmA). The fitted $\varepsilon_{abs}$, $\varepsilon_{rel}$ are $0.075 \times 10^{15}$ molecule cm$^{-2}$ h$^{-1}$ and 0.045. Note that **R** is the
variance of the observation error, the CTM model error and the representation error. Our analyses
showed that the **R** values are dominated by the satellite observation errors ($\sigma_{obs}$). The representation
error can be neglected. We set the small contribution of model errors in **R** to $0.5 \times 10^{15}$ molecule cm$^{-2}$.
To capture the quick changes of NH$_3$ emissions during the fertilizing seasons and give more weight to
satellite observations with high values during the assimilation, we need to reduce their high observation
errors for high values and keep the same observation errors for low values. By fitting NH$_3$ observation
errors ($\sigma_{obs}$) against the observed columns $C$ using all observations in 2020, we find a linear relation:
$\bar{\sigma}_{obs} = \alpha C + b$                 (3)
$\alpha$ is equal to 0.2 and $b$ is equal to $1 \times 10^{15}$ molecule cm$^{-2}$. If the given $\sigma_{obs}$ is larger than $\alpha C + b$, we
use Eq (3) for the observation error in **R**.
We update NH$_3$ emissions only over land since there is almost no NH$_3$ emissions over oceans and seas.
As we mentioned, NH$_3$ reacts with sulfuric and nitric acids from SO$_2$ and NO$_x$ to form PM2.5. The
changes in NO$_x$ and SO$_2$ emissions will affect the concentration and removal of NH$_3$ in the atmosphere.
Inaccurate emissions of NO$_x$ and SO$_2$ will therefore affect the inversion of NH$_3$ emissions. To assess
the sensitivity of NH$_3$ emissions derived with DECSO on NO$_x$ and SO$_2$ emissions, we have run DECSO
with different NO$_x$ and SO$_2$ emissions (default emissions of HTAP v3 and doubling the emissions of
HTAP v3 for SO$_2$ and NO$_x$) as input for the CTM. The results shows that the inversion of NH$_3$ emissions
is not sensitive to the change of SO$_2$ emissions, but it is to NO$_x$ emissions. In Europe, the impact of SO$_2$
emissions on NH$_3$ can be neglected nowadays due to the low SO$_2$ emissions (Luo et al., 2022), which
have been reduced by 80% in 2020 compared to 2005 (EEA, 2023). The sensitivity tests indicate that
up-to-date NO$_x$ emissions are very important for the accurate inversion of NH$_3$ emissions. The monthly
NO$_x$ emissions of HTAP in 2018 and derived with DECSO in 2020 are quite different over the various
countries (Figure S3). In 2020, due to the COVID-19 pandemic, NO$_x$ emissions reduced compared to
other years. van der A et al. (2024) has compared the seasonality of NO$_x$ emissions of DECSO to other
bottom-up inventories and showed individual temporal variability of industrial facilities is derived with
DECSO in Europe, while bottom-up inventories use the same temporal profile per country per sector
and no detailed information of the temporal changes of individual sources. We estimate NH$_3$ and NO$_x$
emissions with DECSO simultaneously (the multi-specie DECSO) from CrIS and TROPOMI on a daily
basis. We use the DECSO-NH3 version to estimate only NH$_3$ emissions from CrIS and use NO$_x$
emissions of HTAP v3 as input for the CTM. Figure 1 shows the difference of monthly NH$_3$ emissions
in three countries (Netherlands, Italy and Greece) derived with the multi-species DECSO version and
the DECSO-NH3 version. The derived NH$_3$ emissions all differ largely (up to ±40%) in winter and less
in summer.




*Figure 1. The relative difference (multi-species DECSO minus DECSO-NH3) of NH₃ emissions between multi-species DECSO and*


*DECSO-NH3. DECSO-NH3 means that only NH₃ emissions are derived with CrIS-NOAA-20. multi-species DECSO means that*
*NH₃ and NOₓ emissions are derived using CrIS-NOAA-20 and TROPOMI observations.*
3. Results
3.1 NH₃ emissions in Europe
We have run the DECSO-parallel version with NH₃ observations from CrIS-NOAA-20 and CrIS-SNPP
respectively to estimate NH₃ emissions over the selected domain of Europe in 2020 (Figure 2). The total
NH₃ emissions over the study domain are 8.0 Tg/year from SNPP and 8.1 Tg /year from NOAA-20.
The spatial distribution of the NH₃ emissions derived from the two satellites agrees well, with small
differences resulting from deviations of the observed NH₃ columns. The spatial distribution of high NH₃
emissions derived from DECSO is similar to that of HTAP, CAMS-REG-ANT and CAMS-GLOB-ANT
but with more local-scale variability and hotspots. The total emissions of DECSO over the European
domain are higher than HTAP (4.2 Tg/year), CAMS-REG-ANT (4.0 Tg/year) and CAMS-GLOB-ANT
(5.9 Tg/year)

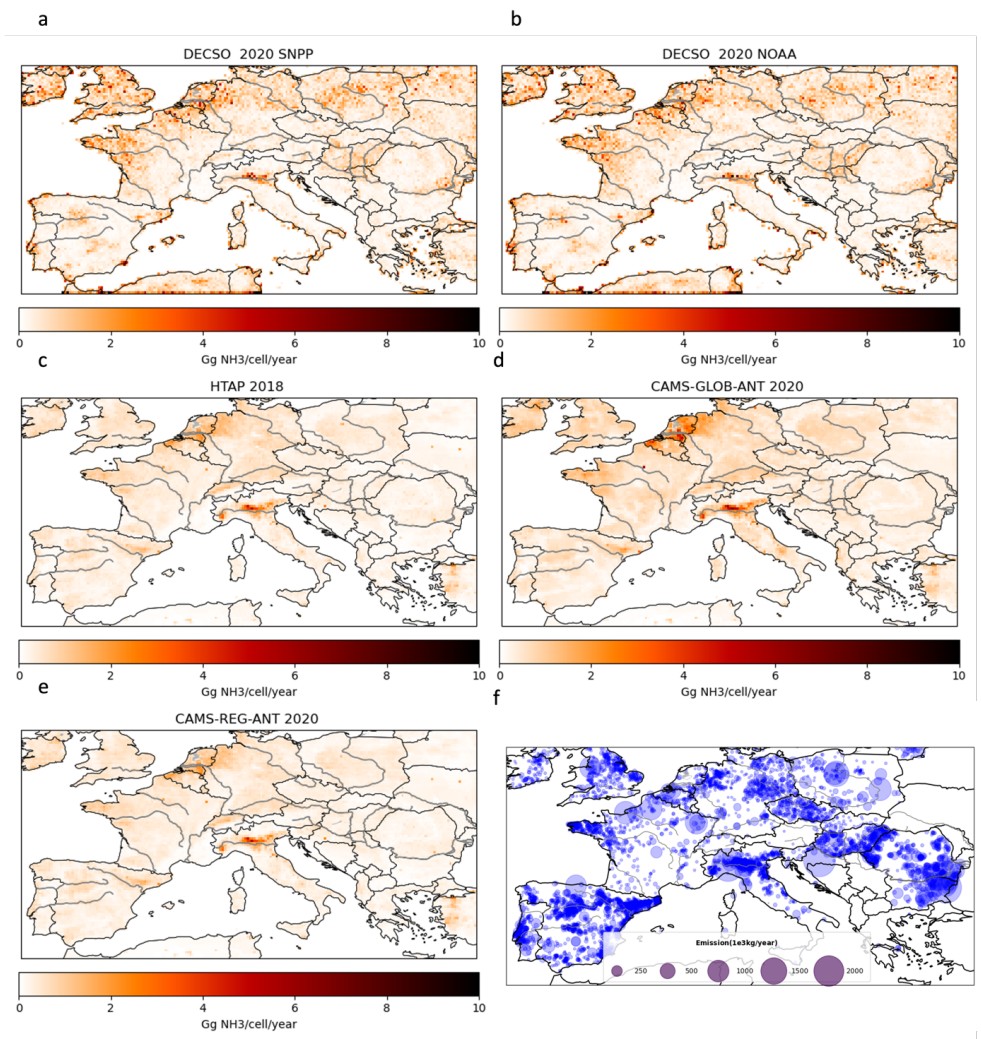

*Figure 2. NH₃ emission maps. NH₃ emissions derived with DECSO from (a) SNPP and (b) NOAA-20 in 2020. NH₃ emissions of*
*(c) HTAP in 2018, (d) CAMS-GLOB-ANT in 2020 (e) CAMS-REG-ANT in 2020. (f) The registered point sources of E-PRTR in*
*2017.*

The locations of high NH₃ emissions, especially in Po-Valley, Spain, Hungary and the east of Romania, shown in DECSO are highly corelated to the registered NH₃ point sources of E-PRTR which are from industrial facilities including livestock facilities but not from fertilizer applications. We see that emissions from the Netherlands are high in DECSO and the bottom-up inventories but are missing in the database of E-PRTR. For the countries in East Europe (e.g. Poland, Hungary, Romania), the NH₃ emissions derived with DECSO are much higher than from the bottom-up inventories. To assess the NH₃ emissions per country, we calculated the country total emissions (see Figure 3). The correlation



coefficients of country totals from DECSO with the bottom-up inventories are all higher than 0.95. In
general, the country totals of NH₃ emissions derived by DECSO from either NOAA-20 or SNPP are
comparable to HTAP, LRTAP, CAMS-REG-ANT and CAMS-GLOB-ANT, with DECSO about 30%
higher. HTAP, LRTAP and CAMS-REG-ANT have very similar emissions per country, while CAMS-
GLOB-ANT shows higher emissions than the other three bottom-up inventories. Because HTAP v3
uses annual emissions from CAMS-REG-ANT for Europe, the only difference between HTAP v3 and
CAMS-REG-ANT is the difference in year. The input of CAMS-REG-ANT is mainly based on LRTAP.
CAMS-GLOB-ANT is based on EDGAR and use different emission activities and factors compared to
the other three bottom-up inventories. In the North part of Europe, for example Netherlands and
Germany, DECSO results show lower NH₃ emissions than CAMS-GLOB-ANT but higher than HTAP,
LRTAP and CAMS-REG-ANT.

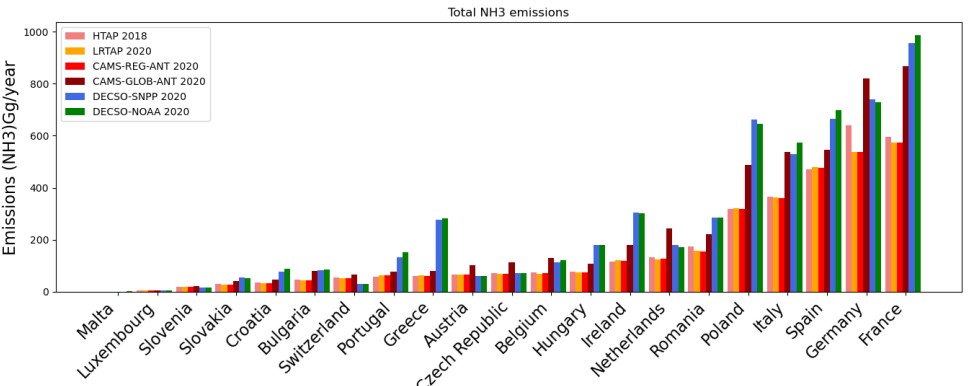


*Figure 3 Country totals of NH₃ emissions (Gg/year) according to database LRTAP in 2018, bottom-up inventories HTAP in*
*2018, CAMS-REG-ANT in 2020, CAMS-GLOB-ANT in 2020 and the DECSO calculations from SNPP and NOAA-20 in 2020.*
To analyze the seasonality of NH₃ emissions derived from DECSO, we compare the monthly emissions
of DECSO with bottom-up inventories. Figure 4 shows the monthly NH₃ emissions from DECSO,
HTAP, CAMS-REG-TEMPO, and CAMS-GLOB-ANT of the Netherlands, Spain, France and Poland.
We see that the seasonal cycle of NH₃ emissions of DECSO are closer to CAMS-GLOB-ANT. HTAP
shows the exact same monthly variability for each country. CAMS-REG-TEMPO shows very similar
monthly patterns to the ones reported by CAMS-GLOB-ANT as they are both using the same method
to derive the temporal profiles for livestock and agricultural soil emissions (Guevara et al., 2021). In
the Netherlands as an example for north Europe, the monthly NH₃ emissions of DECSO are lower than
CAMS-GLOB-ANT but very close to CAMS-REG-ANT. Two peaks of NH₃ emissions show up in
April and August for CAMS emissions. This is also confirmed by the monthly surface concentrations



measured by the MAN network (Figure S4). In Spain and France, the monthly emissions of DECSO
are comparable to CAMS-GLOB-ANT. In the east part of Europe, such as Poland, DECSO estimates
higher emissions. Note that in Spring, when the NH$_3$ emissions are high due to fertilizer applications
on farms, the NH$_3$ emissions derived with DECSO can suffer from a time lag due to insufficient
observations (e.g. due to cloudiness, see Figure S5).

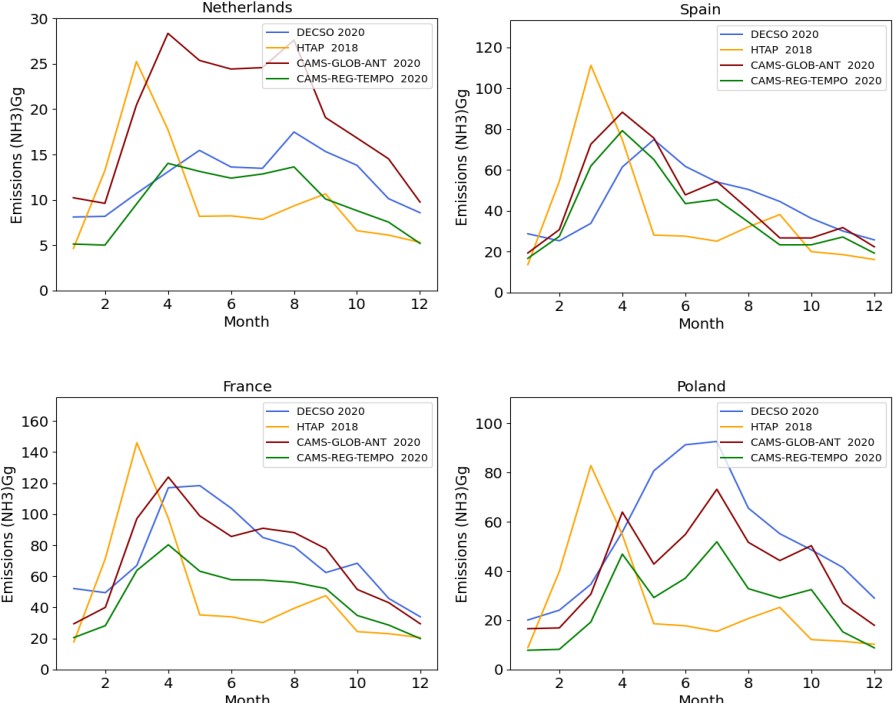


*Figure 4 Monthly NH$_3$ emissions (Gg/month) of DECSO in 2020, HTAP in 2018, CAMS-REG-TEMPO in 2020 and CAMS-GLOB-*
*TEMPO in 2020 for (a) the Netherlands, (b) Spain, (c) France and (d) Poland.*

3.2 Emissions in the Netherlands
On the emission maps of Figure 2, we see that the Netherlands and Po-valley have the highest emission
intensity of NH$_3$. In this section, we focus our analysis on the Netherlands since it has the densest
network for monitoring surface NH$_3$ concentrations and also a detailed emission inventory on a very
high spatial resolution. The total emissions of the Netherlands estimated from the two satellites are very
similar (Figure 3), but the spatial distributions show significant differences (Figure S6). One possible
reason is that about 10% more observations are available from NOAA-20 than SNPP in 2020 (see Figure
S7). The number of valid observations is in general low at high latitudes (Figure S8). More observations
allow the detection of fast changes of NH$_3$ emissions from day to day. By averaging the emissions, the



information from both satellites is combined and improved the quality of the derived emissions due to
a doubling of the number of observations. We use the average of the results of DECSO-SNPP and
DECSO-NOAA-20 to get a better spatial distribution of $NH_3$ emissions derived from satellite
observations.
We compare the total $NH_3$ emissions of DECSO with CAMS-GLOB-ANT, HTAP and official national
$NH_3$ emissions of the Netherlands, which are 148, 230, 122 and 123 Gg/year respectively. DECSO is
lower than CAMS-GLOB-ANT but higher than HTAP and the official $NH_3$ emissions of the
Netherlands. Figure 5 shows the spatial distribution of each inventory in the Netherlands. We see that
DECSO captures the high emission areas and regional distribution over the country. The correlation
coefficients of the spatial distribution of $NH_3$ emissions between DECSO and the national emissions of
the Netherlands, HTAP v3, CAMS-GLOB-ANT are 0.87, 0.87 and 0.88 respectively. At the resolution
of the individual DECSO grid cells, 0.2° × 0.2° grid cell, the emission patterns show differences. This
may be due to uncertainties in the location of the emissions and displacements by up to 0.5° to 1° grid
cell.

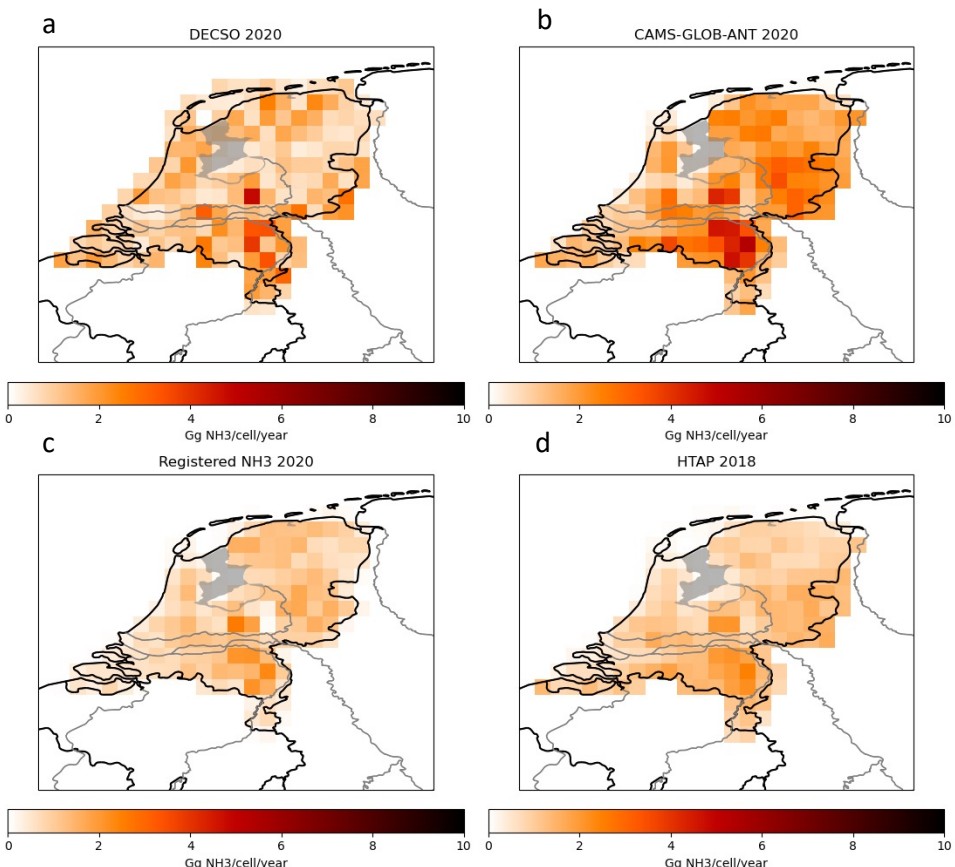


*Figure 5 NH₃ emissions in the Netherlands. (a) The averaged NH₃ emissions derived with DECSO from SNPP and NOAA-20.*

*(b) NH₃ emissions of CAMS-GLOB-ANT in 2020. (c) The official national NH₃ emissions for the Netherlands in 2020 (from*

*emissieregistratie.nl). (d) NH₃ emissions of HTAP in 2018.*

To further assess the DECSO results using in-situ observations from both LML and MAN networks in the Netherlands, we have conducted three runs of CHIMERE for the year 2020 using NH₃ emissions from DECSO in 2020, HTAP in 2018 and CAMS-GLOB-ANT in 2020 over the European domain (same as the setup of DECSO). To compare to the surface NH₃ measurement from the MAN network, we calculate the monthly average of surface NH₃ concentrations from the model simulations. Figure 6 (a-c) shows the scatter plots of monthly NH₃ concentrations of model simulations against observations for the whole year. We see that modelled NH₃ concentrations with the HTAP emissions are underestimated and those with the CAMS-GLOB-ANT emissions are overestimated compared to in-situ observations. The modelled NH₃ concentrations with DECSO emissions have the lowest bias (Figure 7). The performance of model simulations is better in summer months (April to September) than in winter months (October-March). In winter months, few cloud-free satellite observations are





available for the Netherlands. For DECSO, the scatter plot looks more spread out than in summer
months (Figure 6d-i). In summer months, the $NH_3$ concentrations with CAMS-GLOB-ANT are largely
overestimated and with HTAP are largely underestimated, while DECSO has a lower bias compared to
the other two. Note that in the grid cells, the number of stations can vary from 1 to 16. If we select grid
cells with more than 3 sites, DECSO shows better spatial correlation with in-situ observations than for
the other two inventories and the lowest bias (Figure 7 and Table 2).

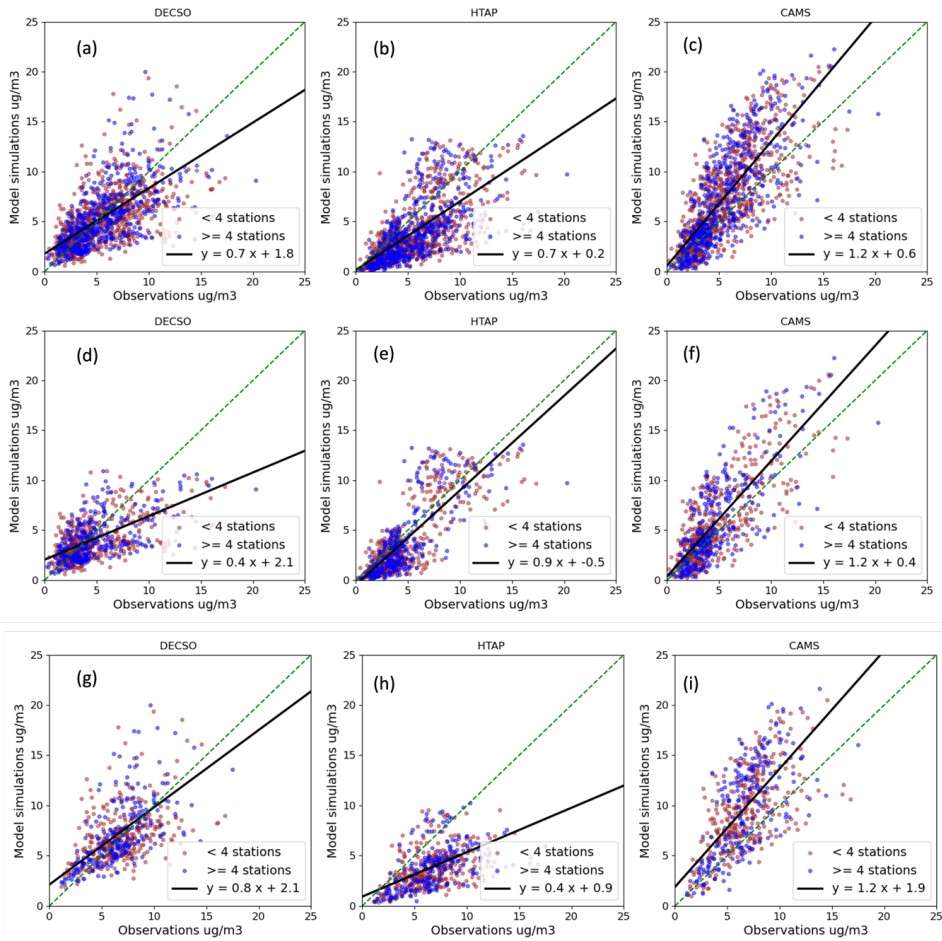


*Figure 6. Scatter plots of observations from the MAN network with $NH_3$ surface concentrations from model simulations with*

*$NH_3$ emissions from DECSO (left column), HTAP (middle column) and CAMS-GLOB-ANT (right column). (a-c) The scatter plot*

*of data for the whole year for all sites. (d-f) The scatter plot of the data in winter months (October to March). (g-i) The*

*scatter plot of the data in summer months (April to September). Each point presents the model grid cells having the in-situ*

*observations. The red dots mean there are less than four in-situ sites in the grid cells. The blue dots mean there are at least*

*four in-situ sites in the grid cell. The fitted black line is for grid cells with at least four in-situ sites.*



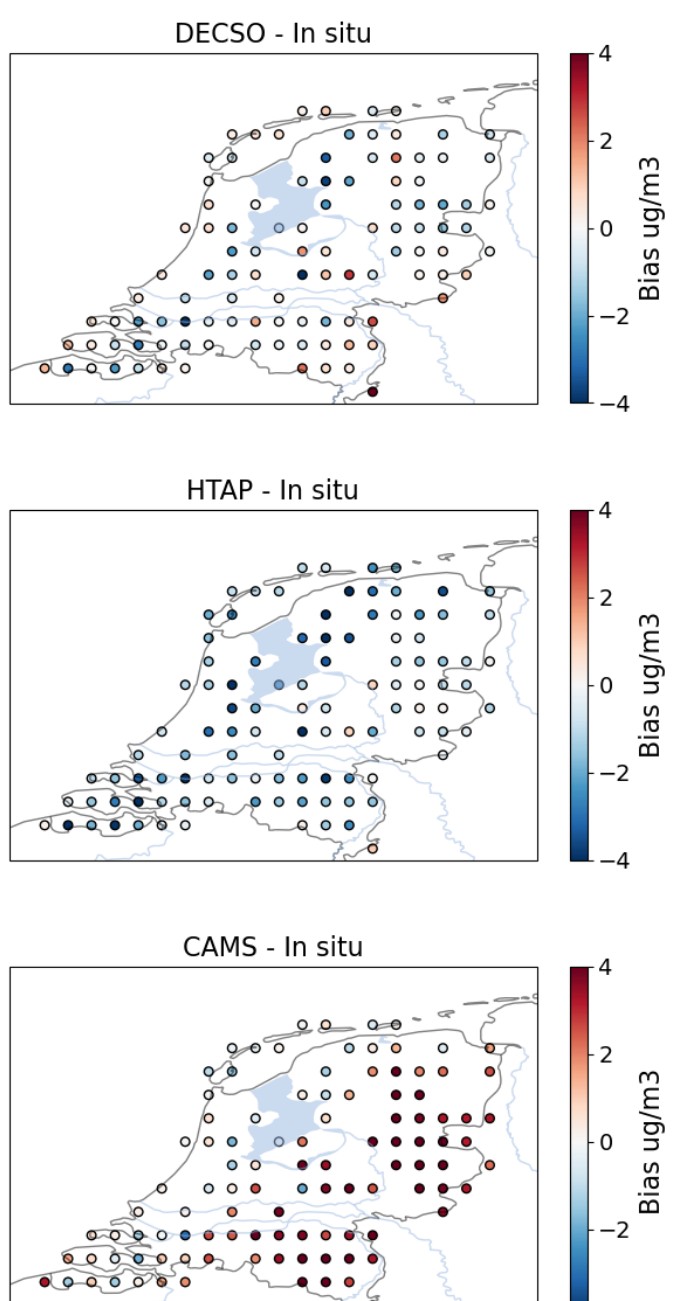

Figure 7. Bias of the model simulated surface concentrations with NH$_3$ emissions from DECSO (left column), HTAP (middle column) and CAMS-GLOB-ANT (right column) compared to the in-situ observations from the MAN network.



*Table 2. The spatial and temporal correlation coefficients and the bias of monthly mean simulated NH$_3$ surface concentration*
*using DECSO, HTAP and CAMS-GLOB-ANT NH$_3$ emissions against observations of the MAN network for grid cells with more*
*than three measurement locations.*

|  | Temporal correlation coefficient | Spatial correlation coefficient | Bias (ug/m$^3$) | RMSE (ug/m$^3$) |
|---|---|---|---|---|
| DECSO | 0.64 | 0.73 | -0.2 | 2.6 |
| HTAP v3 | 0.70 | 0.70 | -1.9 | 3.0 |
| CAMS-GLOB-ANT | 0.82 | 0.70 | -0.3 | 3.8 |



The LML network has six sites measuring surface NH$_3$ concentrations, which are provided every hour.
Since the difference in our model simulations is only due to the monthly input emissions of NH$_3$, we
calculate monthly average NH$_3$ observations for the six sites to compare with the modelled monthly
averaged concentrations. The comparison shows that the model simulations using the DECSO NH$_3$
emissions have similar performance as bottom-up inventories (Figure S9 and S10). The correlations of
modelled monthly NH$_3$ concentration using DECSO and CAMS-GLOB-ANT emissions with the
observations from the LML network are better than that of HTAP, while CAMS-GLOB-ANT has the
lowest bias. Based on these six sites, the comparison shows that the model result using DECSO is very
comparable with that using CAMS-GLOB-ANT.

3.3 Uncertainties and bias of NH$_3$ emissions
One advantage of DECSO is that a standard deviation of derived emissions is also calculated per grid
cell on a daily basis according the Kalman filter equations. As described by van der A et al. (2024), the
derived errors in the emissions are correlated in time linked to the assumption of the persistent emission
forecast model. The autocorrelation effects can be neglected after about one week up to ten days. We
follow the autocorrelation function presented by van der A et al. (2024) to calculate the monthly variance
of NH$_3$ emissions. The monthly variance of NH$_3$ emissions for each grid cell in the study domain varies
from 17% to 58%. For the Netherlands, the precision (random uncertainty) of the monthly emissions is
about 20% and the precision of the annual total is about 5%.



A bias in satellite derived emissions can be introduced due to the linearisation of the averaging kernels
(Sitwell et al., 2022). The CrIS ammonia observations are retrieved in logarithm space together with
logarithmic averaging kernels. As discussed by Sitwell et al. (2022), either using the logarithmic
averaging kernel or the linearized averaging kernel introduces a bias when applying them to the model
simulated profiles. The logarithmic averaging kernels cause problems when the model profiles are zero
at any point in the profile and lead to a positive bias in emission estimates. Linearized averaging kernels
may introduce a negative bias in emissions when there is a large difference between the model profile
and the a priori profile used in the retrieval.

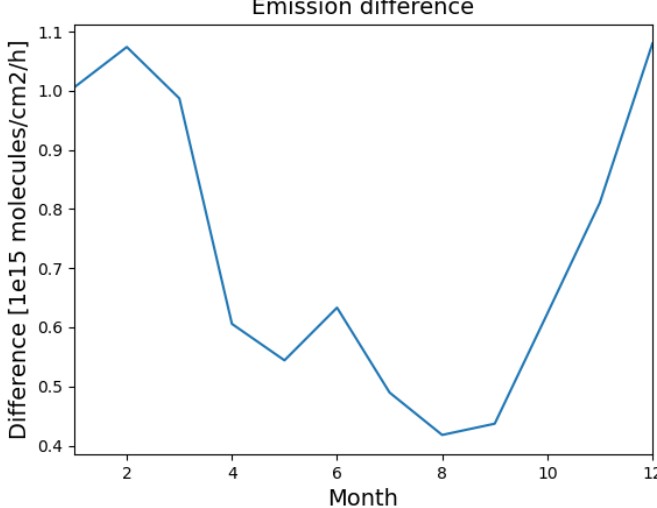


*Figure 8. The absolute change of monthly $NH_3$ emissions (molecule/cm²/h) if there is a positive bias of $5 \times 10^{15}$*
*molecule/cm² of each $NH_3$ column observation.*
To assess how the biases in satellite $NH_3$ observations affect emissions derived by DECSO, we have
done two simple bias tests. For the first test, the $NH_3$ columns of CrIS on NOAA-20 are increased by
20%, a positive relative bias for the satellite observations. The annual emissions of $NH_3$ with the
introduced bias increase by 27% for the European domain. It seems that the introduced bias has a higher
impact on emissions in winter than in summer. The relative bias on emissions can be as high as 50% in
winter. The change of emissions in summer becomes even negative probably because $NH_3$ column
concentrations can show a large variation from day to day. When the $NH_3$ columns are very high on
one day and next drop to a very low value, the absolute change in concentration is larger than the
original situation without introduced bias. This will lead to a larger decrease in the updated emissions
and can result in a negative change of emissions.



For the second test, an absolute bias of $5 \times 10^{15}$ molecule/cm$^2$ is added to each NH$_3$ column observation
of CrIS on NOAA-20. Figure 8 shows the increase of NH$_3$ emissions caused by the absolute bias
introduced in the satellite observations. We see that the increase is doubled in winter compared to
summer, because the lifetime in winter is longer than in summer. The averaged effective lifetime
calculated with DECSO is about 10 hours in winter and 5 hours in summer. With the same bias of NH$_3$
columns, the impact on emissions is larger in winter than in summer.

## 4. Conclusions
To derive NH$_3$ emissions from satellite data, we presented an updated version of the DECSO algorithm
with specific settings for NH$_3$. Together with the improved the DECSO version for NOx of (van der A
et al., 2024), we have the multi-species DECSO version to update NO$_x$ and NH$_3$ emissions
simultaneously. In general, the removal of NH$_3$ in the atmosphere is affected by the amount of NO$_x$ and
SO$_2$ emissions. For the study domain of Europe, our sensitivity study shows that the influence of
changes in NO$_x$ emissions need to be considered in the inversion of NH$_3$ emissions in DECSO. The
impact of SO$_2$ emissions is very small and can be neglected since the SO$_2$ emissions are usually low in
Europe. Thus, to derive NH$_3$ emissions and to analyze the seasonal cycle and trend of NH$_3$ emissions
from satellite observations over Europe, it is recommended to include updated NO$_x$ emissions in the
inversion calculation of NH$_3$ emissions in DECSO. For regions with high SO$_2$ emissions, it is necessary
to consider if the SO$_2$ emissions are changing rapidly and are up-to-date in the inversion.
The error covariances of the updated daily NH$_3$ emissions per grid cell are provided during the
calculation in DECSO. Considering the autocorrelations introduced by the assumption of the
persistency emission model, the calculated monthly error on NH$_3$ emissions for each grid cell in the
study domain varies from 17% to 58%. The yearly error per grid cell is about 5 ~ 15%. The sensitivity
tests for retrieval biases shows that with an introduced constant relative and absolute bias in NH$_3$
retrievals, the resulting bias in emissions derived with DECSO shows a seasonal variability with a peak
in winter. This means the algorithm is more sensitive to a bias in the observations during wintertime.
The total NH$_3$ emissions in our European domain derived from NH$_3$ observations of SNPP and NOAA-
20 are 8.0 Tg/year and 8.1 Tg/year respectively with a precision of about 0.2 %. The difference in
country total emissions derived from the two satellites is very small. However, the details of the spatial
distribution of emissions derived from both satellites are different over the north part of the domain,
such as the Netherlands. This may be due to the varying number of observations per region per year
from the two satellites. An average of the emissions derived from both satellites leads to an improved
spatial distribution compared to the emissions from the individual satellite. The spatial distribution of
derived NH$_3$ emissions is similar to the bottom-up inventories, but DECSO emissions are in general
higher. The annual total emissions derived by DECSO for the whole domain is larger than the bottom-



up inventories (LRTAP, HTAP, CAMS-REG-ANT and CAMS-GLOB-ANT). The comparison of
country total emissions shows that DECSO gives higher $NH_3$ emissions for the countries in East Europe
than the bottom-up inventories. In addition, DECSO results show higher sources in Spain, Hungary and
the east of Romania. This is in line with the registered point sources of E-PRTR. The seasonal cycle of
the emissions of DECSO are comparable to CAMS-GLOB-ANT, while HTAP uses the same seasonal
cycle for each country in Europe.
For the Netherlands, model simulations using $NH_3$ emissions from DECSO, HTAP and CAMS-GLOB-
ANT are compared to in-situ observations from the MAN and LML networks. In general, the simulation
using DECSO emissions has a lower bias, but also a lower temporal correlation compared to CAM-
GLOB-ANT. The performance of model simulations with DECSO is better in summer than in winter.
Both the bias and spatial correlation between model simulations using DECSO emissions and the MAN
in-situ observations are higher than CAMS-GLOB-ANT for grid cells including more than three
measurement sites. We conclude that satellite-derived emissions derived with DECSO show a
comparable temporal and spatial distribution as bottom-up inventories. The emissions derived from
satellite observations can provide fully independent information on emissions for verifying the bottom-
up inventories. With the global coverage of satellite observations, DECSO can be easily applied to
different regions. After validation of DECSO over regions like Europe, where there is sufficient
information of emissions, the added value of DECSO for deriving $NH_3$ emissions is to provide $NH_3$
emissions over regions with limited local information of $NH_3$ emissions.

Data
The CrIS $NH_3$ data v1.6.4 of SNPP and NOAA-20 created by Environment and Climate Change Canada
are currently publicly available upon request (mark.shephard@canada.ca) at
https://hpfx.collab.science.gc.ca/~mas001/satellite_ext/cris/snpp/nh3/v1_6_4.
The TROPOMI NO2 data version 2.4 are available via the Copernicus website
https://dataspace.copernicus.eu/ and via the TEMIS website
https://www.temis.nl/airpollution/no2.php.
The $NH_3$ and $NO_x$ emissions of DECSO v6.3 are available on the GlobEmission website
https://www.temis.nl/emissions/data.php.
HTAP v3 dataset are available on https://edgar.jrc.ec.europa.eu/dataset_htap_v3
The European emissions data sets for countries NEC, LRTAP and large facilities E-PRTR are available
on the website https://www.eea.europa.eu/en/analysis of the EEA.



The CAMS databases CAMS-REG-ANT v5.1, CAMS-GLOB-ANT, CAMS-TEMPO are available on the
ECCAD website https://permalink.aeris-data.fr.
The NH$_3$ observation data from the LML network are available on the RIVM website
https://data.rivm.nl/data/luchtmeetnet/.
The NH$_3$ observation data from the MAN network are available at https://man.rivm.nl.
The Dutch registered NH$_3$ emissions are available at https://data.emissieregistratie.nl/export


Author contribution
JD developed the inversion algorithm of NH$_3$, performed all emission inversions, conducted the
analysis and wrote the manuscript. RA and JD made the improvement of the inversion algorithm of
NOx. HE developed the superobservation code. ED  provided the code for a linearization of the
averaging kernels of CrIS. MS provided the CrIS data. RWK provided the NH$_3$ observation data from the
MAN and LML networks. MGV provided the CAMS-TEMPO profiles. LT provided suggestions during the
research. All authors contributed to the reviewing and editing of the manuscript.

Competing interests
The authors declare that they have no competing interests.

Acknowledgments
This the work was financed by the Sentinel EO-based Emission and Deposition Service (SEEDS, Grant
ID 101004318) project that has received funding from the European Union's Horizon 2020 research
and innovation programme. Part of the work was funded by the Nationaal Kennisprogramma Stikstof
(NKS) of the Dutch Ministry of Agriculture, Nature and Food Quality.

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
