# Peer review of "Ammonia emission estimates using CrIS satellite observations over"

_EGUsphere, 2024_

## Author Comment (AC1)

Response to review

We thank the reviewers for their time spent to help us improving the paper with their useful suggestions. Below we provide a point-by-point response to the reviews and descriptions how we have addressed the suggestions. The original review text is indicated in blue and the response in black. New text appearing in the revised paper is indicated by underline.

Referee #1,

General comments:

Ding et al. presented a comparison of $NH_3$ emissions in Europe derived from the CrIS satellite instruments and existing emission inventories. The authors included $NO_2$ observations from the TROPOMI instrument in the emission calculation as $NH_3$ emissions were found to be sensitive to $NO_X$. They also compared satellite and inventory derived $NH_3$ surface concentrations to the Dutch monitoring networks and found those using satellites had least bias against in-situ observations. This work makes a good example of why we need accurate emission data for policy interventions as bottom-up inventories often fall behind real-world changes. In this regard, satellites that offer daily global coverage are an excellent tool. Overall, I am curious as to why the authors only studied 2020 despite that CrIS and TROPOMI have data records that can be jointly studied beginning in 2018 (first year of TROPOMI operational product) until current day. We know 2020 was a special year in recent decades as anthropogenic activities and emissions were largely altered by the pandemic. As the authors mentioned, $NO_x$ emissions decreased in 2020. Meanwhile, $NH_3$ was found to have an "anomalous increase" during the lockdowns (Kuttippurath et al., 2024). It is not surprising that inventories did not capture these sudden changes as observations (satellite and in-situ) did. In fact, the authors clearly stated one of the inventories was the 2018 data, which does not make a very fair comparison. Since DECSO is not computationally demanding to run, I would be curious to see at least one more year of results, otherwise it is difficult to comment on the general applicability of DECSO.

We choose the year 2020 because this is the only year that has a full year overlap of CrIS NH3 observations of the two satellites SNPP and NOAA-20. For the SNPP satellite, there are no observations from Apr. to Aug. in 2019 and its $NH_3$ retrieval product ends in May 2021. The $NH_3$ retrieval product of NOAA-20 starts from 2019 March. By comparing the $NH_3$ emissions derived from the two satellite observations, we found that there is no significant bias between them. For the next study, we can use the observations from the two satellites to perform a trend study of $NH_3$ emissions using DECSO.

Meanwhile, we have run DECSO with $NH_3$ observations of NOAA-20 for 2021 and 2022 as well. These resulting NH3 emissions for 2020 to 2022 are presented in the file of supplementary information. (Figure S11).

We added the following text in section 2.1.1.

"The $NH_3$ profiles are retrieved on 14 pressure levels with the peak sensitivity of CrIS between 900 and 700 hPa (Shephard et al., 2020). For SNPP, the retrieval products start from 2011 and ends in May 2021 with missing data from April to August in 2019. The $NH_3$ retrieval product of NOAA-20 starts from March 2019. We use the version 1.6.4 retrieval products of CrIS on both SNPP and NOAA-20 from September 2019 to December 2020…"

In section 3.1, we further clarify:

"We have run the multi-species DECSO version with $NH_3$ observations from CrIS-NOAA-20 and CrIS-SNPP respectively to estimate $NH_3$ emissions over the selected domain of Europe in 2020 (**Error! Reference source not found.**), which is the only year with a full year overlap of $NH_3$ observations for these two satellites. "

[Figure]

*Figure S11. Country totals of $NH_3$ emissions (Gg/year) derived from DECSO calculations from SNPP in 2020 and NOAA-20 in 2020, 2021 and 2022.*

Specific comments:

Line 28: Is the 0.2% precision the uncertainty of derived emissions or the agreement between the two satellites? Also please include the uncertainty of other emission estimates, such as Line 305-306.

0.2% is only the precision of the derived emissions excluding biases over the whole year for the whole domain, which is about 5% to 17 % per grid cell per year. One advantage of using the Kalman filter for the inversion of emissions is that it also gives error estimates for each grid cell every day. The monthly variance of $NH_3$ emissions for each grid cell in the study domain varies from 17% to 58%. For the Netherlands, the precision (random uncertainty) of the monthly emissions is about 20% and the precision of the annual total is about 5%. This has been discussed in section 3.3. For other inventories, the uncertainty numbers per grid cell are not given.

To make it less confusing, we have changed the text in the manuscript:

The total $NH_3$ emissions derived from observations are about 8 Tg/year with a precision of about 5-17 % per grid cell per year over the European domain [-10° ~30° E, 35° ~ 55° N].

Line 83: You should clarify that JPSS-1 was renamed as NOAA-20, which is the name being used later in the text. Also clarify JPSS-2 (NOAA-21) was launched much later and therefore not used in the study.

We have clarified that JPSS-1 and JPSS-2 are NOAA-20 and NOAA-21 in the text. Line 83 briefly introduces the background of satellite instruments measuring NH3. In the section 2.1, it is specifed that we use $NH_3$ observations from SNPP and NOAA-20.

Line 174: Can you expand on what you mean by superobservations and how is it different from the original observations?

Super-observations are created by averaging multiple TROPOMI observations over the grid cells of the DECSO-CHIMERE system. This approach has several advantages: it reduces the computational costs and allows a treatment of spatial correlations and representativity errors in the TROPOMI observations. The super-observation error takes into account spatial correlations between individual TROPOMI observations as well as representativity errors in the case of incomplete coverage. The use of super-observations improves the signal-to-noise ratio, and it reduces the calculation time of DECSO.

We have added some explanation also in the text:

$NO_2$ columns are converted into 'super-observations' representing the integrated average (Boersma et al., 2016; Rijsdijk et al., 2024) over the 0.2° × 0.2° grid cells. The super-observation error takes into account spatial correlations between individual TROPOMI observations and representativity errors in the case of incomplete coverage. In this paper, the super-observations are calculated for the $NO_2$ columns from surface till about 700hPa where the $NO_2$ concentrations are most related to surface emissions. The signal-to-noise ratio and calculation time of DECSO are improved by using super-observations.

Line 187: How large is the uncertainty in the in-situ data? This is an important part that should be discussed since you are using in-situ data as the reference for comparing satellites and inventories.

Swart et al. (2023) showed that the median random error of miniDOAS hourly measurements for $NH_3$ is about 23%. For the MAN network, the passive sample have the standard uncertainty for a single monthly mean measurement is 22%.

We added this in the text:

"The uncertainty in $NH_3$ concentrations measured with individual passive samples is large (22% for a single monthly measurement) and the measurements are calibrated monthly against the high-quality measurements (about 20% for an hourly measurement) from the LML network to enhance the accuracy."

Line 298: Is this DECSO-parallel version using the setting of multi-species DECSO or DECSO-NH3 only?

Here it is the multi-species DECSO. We have changed this in the manuscript.

Line 315: Do you know if the underestimation of emissions in East Europe by inventories is due to inaccurate information collected in those countries or other reasons? A little more insight on this would be helpful.

It is very difficult to find how accurate the emission information per country is for the bottom-up inventories. This study shows the spatial difference in $NH_3$ emissions between the satellite and the bottom-up approaches. The reason of the difference should be discussed in the emission research community. Since we could not find any detailed information for $NH_3$ emissions over East Europe, we don't know what the reason of this underestimation is.

Line 373: Can you explain more specifically what caused the displacements of emissions? Is it intrinsic to the data, or does it have something to do with the way you aggregate the emissions (Line 213)?

The derived emissions are updated by addition, not by scaling the existing emissions. In DECSO, the sensitivity of concentrations to emissions is calculated by using trajectory analyses to account for transport away from the source. If the real emission source is at the edge of one grid cell and observed by different pixels each day, the emissions can be allocated to the neighbouring grid cells. As a result, the derived emissions are spread to adjacent grid cells. This is also explained in the paper of van der A et al. (2024).

We have adapted the text:

"At the resolution of the individual DECSO grid cells, $0.2° \times 0.2°$ grid cell, the emission patterns show differences. This may be due to uncertainties in the location of the emissions and displacements by up to one grid cell, similar as for NOx emissions (van der A et al., 2024). For example, the emission sources at the edge of grid cells can be spread to the neighbouring grid cells. "

Line 387: Please explain how you define and compare the biases here, on Line 392, and in Table 2 (assuming they mean the same concept). I can see from Figure 7 that HTAP is underestimating and CAMS is overestimating. However DECSO also has many extreme values indicating it can overshoot either way, so I'm not sure if it really is the better option, especially if you look at Table 2 the spatial correlations and biases of DECSO and CAMS aren't that different (with CAMS having a much better temporal correlation). And please include the uncertainties of these numbers too.

The modelled concentration using different emissions are compared to the in-situ data of the MAN network (modelled concentrations minus in-situ observations). The uncertainties of emissions from CAMS and HTAP are not provided. In-situ uncertainties of MAN is 22% and we have added this in section 2.2. CAMS is currently one of the best bottom-up inventories. We use the latest CAMS as a benmark for deriving emissions from satellite observations. The correlation and biases of DECSO and CAMS are not that different. This means two independent approaches give similar values for NH3 emissions. This means for the regions without good information on emissions, we can use the emissions deriving from satellite observations.

We make it clear in the text:

The modelled $NH_3$ concentrations with DECSO emissions have the lowest absolute bias (modelled concentration minus in-situ observations of the MAN network) (Figure 7).

Line 417: I would include the correlation coefficients and biases here or show them on Figure S9.

We added table S1 to show the correlation coefficients and bias.

*Table S1. The correlation coefficients and the bias of monthly mean simulated NH₃ surface concentrations using DECSO, HTAP and CAMS-GLOB-ANT NH₃ emissions against observations of the LML network according to Figure S9.*

|  | Correlation coefficient | Bias (ug/m³) | RMSE (ug/m³) |
|---|---|---|---|
| DECSO | 0.63 | -3.5 | 6.4 |
| HTAP v3 | 0.58 | -5.0 | 7.6 |
| CAMS-GLOB-ANT | 0.68 | -0.69 | 5.5 |

Line 480: Given that your derived emissions (8 Tg/year) are almost twice as high as inventories (4.0-5.9 Tg/year), how much of this difference do you think may be attributed to the fact that the inversion is solely based on satellite overpasses in the afternoon when NH₃ concentration is usually higher than rest of the day (in other words, not incorporating diurnal changes)?

In the inversion method, the concentrations are calculated from the emissions by a chemical transport model (CTM) and compared to satellite observations. The diurnal cycle is considered in the CTM, which is the same as assumed in the CAMS inventories. Therefore, the emissions mentioned in the paper are already corrected for this diurnal cycle.

Technical corrections:

Line 18: "2.5" in PM2.5 should be subscripted throughout the text.

We have corrected these.

Line 54: Studies show

We have revised it.

Line 122: to validate emissions

We have revised it.

Line 151: The NH₃ profiles

We have revised it.

Line 268: there are almost no

We have revised it.

Line 331: Figure 3 caption says LRTAP 2018 but figure legend says LRTAP 2020.

We have used LRTAP 2020 for comparison. We have revised the caption.

Line 336: is closer to

We have revised it.

Line 345: Spring does not need to be capitalized here.

We have revised it.

Line 462: redundant "of"

We have revised it.

References:

Kuttippurath, J., Patel, V. K., Kashyap, R., Singh, A., & Clerbaux, C. (2024). Anomalous increase in global atmospheric ammonia during COVID-19 lockdown: Need policies to curb agricultural emissions. Journal of Cleaner Production, 434, 140424. https://doi.org/10.1016/j.jclepro.2023.140424

Referee #2

This MS deals with the ammonia emission estimates for Europe using satellite observations. As there are not many studies like this, the MS has new information and thus, can be considered for publication. However, there are certain things to be clarified before the MS can be accepted.

Minor:

1. Neither the Abstract nor Conclusions provides a clear message about this study. What is the main conclusion from this study? Just the comparison of different estimates? How we can improve the emission estimates? What are current uncertainties? What uncertainty is addressed in this study? This has to be clearly mentioned in the Abstract and Conclusion, and the problem must be properly discussed in the Introduction too. Now this is just a comparison paper, as you stated in the last sentence of the Abstract.

In this study, we have shown that montly $NH_3$ emissions on 20 km resolution can be derived by using satellite observations. The DECSO inversion showed a highly variable $NH_3$ seasonal cycle over countries and years, which is in general ignored in the current bottom-up inventory. This monthly profile can provide extra information to validate and understand temporal profiles of bottom-up inventories. We implemented DECSO to estimate $NO_x$ and $NH_3$ emissions simultaneously and show the importance of the impact of $NO_x$ emissions on the inversion of $NH_3$ emissions from satellite observations. This indicate that for the trend analysis of $NH_3$ emissions using satellite observations in Europe, it is necessary to consider the trend of $NO_x$ emissions. The small difference of country total $NH_3$ emissions derived from SNPP and NOAA20 observations show that we can use observations of the two satellite to perform long term trend study at least over the country scale. As we mentioned in our conclusion, the emissions derived from satellite observations can provide fully independent information on emissions for verifying the bottom-up inventories. With the global coverage of satellite observations, DECSO can be easily applied to different regions. After validation of DECSO over regions like Europe, where there is sufficient information of emissions, the added value of DECSO for deriving $NH_3$ emissions is to provide $NH_3$ emissions over regions with limited local information of $NH_3$ emissions.

To make it more clear to the reader, we have revised the abstract and conclusion:

Abstract:

"…. observations indicates a consistency in terms of magnitude on the country totals, the results also being comparable regarding the temporal and spatial distributions. The validation of DECSO over Europe implies that we can use DECSO to quickly derive fairly good monthly emissions of $NH_3$ over regions with limited local information of $NH_3$ emissions. "

Conclusion (line 504):

"The analysis indicates that DECSO can be used to estimate NH$_3$ over a long period for the trend study. The retrieval product of NH$_3$ from SNPP ends in May 2021. Because of the insignificant differences in NH$_3$ emissions derived from the two satellites for the overlap year 2020, the trends analysis can be continued by using the NH$_3$ data from NOAA-20 (Figure S11. We have shown the importance of the impact of NO$_x$ emissions on the inversion of NH$_3$ emissions. Since the NO$_x$ emissions derived from TROPOMI have good agreement with CAMS-REG-ANT, as shown by (van der A et al., 2024). the NO$_x$ emissions from CAMS-REG-ANT can be used for the years before 2019 in trend studies of NH$_3$ emissions over Europe. "

[Figure]

*Figure S11. Country totals of NH$_3$ emissions (Gg/year) derived from DECSO calculations from SNPP in 2020 and NOAA-20 in 2020, 2021 and 2022.*

2. Please state why the year 2020 is selected for this particular study.

We chose the year 2020 because this is the only year there is a full year overlap of NH$_3$ observations for these two satellites. For the SNPP satellite, there is no observations from Apr. to Aug. in 2019 and its NH$_3$ retrieval product ends in May 2021. The NH$_3$ retrieval product of NOAA20 starts from 2019 March. We want to check if there is a significant deviation between the emissions derived from both satellites. By comparing the NH$_3$ emissions derived from the two satellite observations, we know there is no significant bias between them. For the future studies, we can use the observations from the two satellites to perform a trend study of NH$_3$ emissions using DECSO.

In section 3.1, we clarified it:

"We have run the multi-species DECSO version with NH$_3$ observations from CrIS-NOAA-20 and CrIS-SNPP respectively to estimate NH$_3$ emissions over the selected domain of Europe in 2020 (**Error! Reference source not found.**), which is the only year with a full year overlap of NH$_3$ observations for these two satellites."

3. When you add NOx, how much the emission estimates improved in percent? Please state this.

It is hard to just give one number to describe the improvement, which depends on the change of NOx emission locally. This mean the improvement is different per grid cell per day. In section 2.4 and Figure 1, we show the relative difference of monthly NH$_3$ emissions in three countries by adding NOx as example. The derived NH$_3$ emissions all differ largely (up to ±40%) in winter. The improvement in percentage is higher in winter than in summer.

4. There is a clear seasonal difference in emission estimates. This suggests that the meteorology plays a big part in these calculations. So if you change meteorological input, how much will be the difference? If you consider the uncertainties in meteorological data, what would be the uncertainty or bias in the estimated emissions?

Meteorological parameters such temperature, relative humidity have high impact on NH$_3$ and NO$_2$ concentrations and emissions. In the inversion calculation this influence is shown in the effective lifetime of NH$_3$. The lifetime are higher in summer (10 hour) and lower (5 hour) in winter. The uncertainties of meteorological data are part of the uncertainties of the CTM, which are considered in the inversion calculation (kalman filter). If there is a large bias of the meteorological data, this will lead to bias in the emission inversion. In this study, we use the analysis data of ECMWF, which has a very limited error for the resolution that we are using..

5. Conclusion is almost two pages, but I do not see that much discussion in the paper. So please make it short. A 350-word Conclusion would be much more effective than this.

We change the section "conclusion" to "discussion and conclusion" since the discussions are indeed described in that section and we have added more discussion as mentioned earlier

6. I find some language issues in the MS, please pay attention to that.

We have checked and revised the language issues in the MS.

technical:

L19: affect

changed

L26: limited daily observation

 We have changed it to "the limited number of daily observations".

L28: be specific about the region you mentioned here

We have added the coordinates of the region.

It is not a citation, but the name of a network of nature protection areas in the territory of the European Union . We have made it clear in the manuscript:

"…, because it was inadequate for the protection of vulnerable nature areas (named Natura 2000). "

We have change the sentence into:

Ge et al. (2020) summarized the key factors of agricultural $NH_3$ emissions: local agricultural practices, manure and fertilizer application including type, amount and method, animal species, housing, manure storage, meteorological conditions, soil properties, and regulations of agricultural practice.

Thank you for pointing out this paper. We added this paper as a reference in line 270.

L301: How small is this "small"? Please specify the value

The root mean square difference of $NH_3$ emissions between DECSO-SNPP and DECSO-NOAA-20 is about 0.0067 Gg/ $NH_3$/Cell/year. The average NH3 emissions is 0.55 Gg/$NH_3$/cell/year. The relative root mean square difference is about 1.2%. We specified the value in the paper.

L385, L388: Please state the reasons for the differences

The difference of input emissions leads to different results of modelled concentrations

L410: If you look at the table, the comparison shows better for DESCO, but Figure 7 shows the least bias for HTAP?

L420: Figure 7 shows the smallest bias for HTAP

Figure 7 shows a negative bias of HTAP and a postive bias of CAMS. On average, DECSO shows the lowest absolute bias. The colorscale of this figure is probably not very clear. We have re-made the figure.

[Figure]

*Figure 7. Bias of the model simulated surface concentrations with NH₃ emissions from DECSO (left column), HTAP (middle column) and CAMS-GLOB-ANT (right column) compared to the in-situ observations from the MAN network.*

The DECSO algorithm we present here can be applied to any region.

Added references:

Boersma, K. F., Vinken, G. C. M., and Eskes, H. J.: Representativeness errors in comparing chemistry transport and chemistry climate models with satellite UV–Vis tropospheric column retrievals, Geosci. Model Dev., 9, 875-898, 10.5194/gmd-9-875-2016, 2016.

Ge, X., Schaap, M., Kranenburg, R., Segers, A., Reinds, G. J., Kros, H., and de Vries, W.: Modeling atmospheric ammonia using agricultural emissions with improved spatial variability and temporal dynamics, Atmos. Chem. Phys., 20, 16055-16087, 10.5194/acp-20-16055-2020, 2020.

Rijsdijk, P., Eskes, H., Dingemans, A., Boersma, F., Sekiya, T., Miyazaki, K., and Houweling, S.: Quantifying uncertainties of satellite NO2 superobservations for data assimilation and model evaluation, EGUsphere, 2024, 1-42, 10.5194/egusphere-2024-632, 2024.

Shephard, M. W., Dammers, E., Cady-Pereira, K. E., Kharol, S. K., Thompson, J., Gainariu-Matz, Y., Zhang, J., McLinden, C. A., Kovachik, A., Moran, M., Bittman, S., Sioris, C. E., Griffin, D., Alvarado, M. J., Lonsdale, C., Savic-Jovcic, V., and Zheng, Q.: Ammonia measurements from space with the Cross-track Infrared Sounder: characteristics and applications, Atmos. Chem. Phys., 20, 2277-2302, 10.5194/acp-20-2277-2020, 2020.

Swart, D., Zhang, J., van der Graaf, S., Rutledge-Jonker, S., Hensen, A., Berkhout, S., Wintjen, P., van der Hoff, R., Haaima, M., Frumau, A., van den Bulk, P., Schulte, R., van Zanten, M., and van Goethem, T.: Field comparison of two novel open-path instruments that measure dry deposition and emission of ammonia using flux-gradient and eddy covariance methods, Atmos. Meas. Tech., 16, 529-546, 10.5194/amt-16-529-2023, 2023.

van der A, R. J., Ding, J., and Eskes, H.: Monitoring European anthropogenic NOx emissions from space, Atmos. Chem. Phys., 24, 7523-7534, 10.5194/acp-24-7523-2024, 2024.

---

## Author Response (AR2)

We thank the editor for the correction. The original text is indicated in italic blue font and the response in regular black font.

Table S1: please correct units to μg/m³

This has been changed

Data section:

Doi link in line 533 not working, doi not found

We revised the doi here.

line 540: please check availability, website was inaccessible when I checked, resulting in failure to load due to time-out

We have provided the correct link.

Lines 543 and 544: are there English version of the websites that could be linked?
Unfortunately there are no English websites of these datasets.

Line 540, please provide a more specific link, this is too general

We have provided specific links to these datasets.